# WHEN SCALE IS FIXED: REVISITING PRE-TRAINING INDICATORS FOR LLM FINE-TUNING PERFORMANCE

## ABSTRACT

While pre-trained available metrics, such as perplexity, correlates well with model performance at scaling-laws studies, their predictive capacities at a fixed model size remains unclear, hindering effective model selection and development. To address this gap, we formulate the task of selecting pretraining checkpoints to maximize downstream fine-tuning performance as a pairwise classification problem: predicting which of two LLMs, differing in their pre-training, will perform better after supervised fine-tuning (SFT). We construct a dataset using 50 1B parameter LLM variants with systematically varied pre-training configurations, e.g., objectives or data, and evaluate them on diverse downstream tasks after SFT. We first conduct a study and demonstrate that the conventional perplexity is a misleading indicator. As such, we introduce novel unsupervised and supervised proxy metrics derived from pre-training that successfully reduce the relative performance prediction error rate by over 50%. Despite the inherent complexity of this task, we demonstrate the practical utility of our proposed proxies in specific scenarios, paving the way for more efficient design of pre-training schemes optimized for various downstream tasks.

## 1 INTRODUCTION

Large Language Models (LLMs) (Google et al., 2024; OpenAI, 2023; Chowdhery et al., 2023; Grattafiori et al., 2024) are central to contemporary NLP, powering systems like Chatbots and specialized assistants. They are typically employed via few-shot prompting or task-specific fine-tuning. Despite the accessibility afforded by prompting, fine-tuning on downstream tasks is often indispensable for optimal model performance, particularly within specific application domains or when utilizing private data (Singhal et al., 2025; Lee et al., 2024a; Lai et al., 2023).

While LLMs demonstrably improve on supervised fine-tuning (SFT) tasks with increasing scale (Zhang et al., 2024; Isik et al., 2025), the substantial costs associated with larger models strongly motivate performance optimization at a fixed size. These efforts often concentrate on refining pre-training elements, such as data compositions (Shen et al., 2024; Penedo et al., 2024) or training objectives (Raffel et al., 2020; Tay et al., 2023a;b). This context underscores a critical need: the ability to reliably forecast the post-SFT performance of same-size LLM variants using only indicators available during pre-training. Although metrics like perplexity correlate well with scaling-driven performance gains (lower perplexity generally corresponds to better few-shot (Grattafiori et al., 2024) and fine-tuning (Isik et al., 2025) results as model size expands), their predictive efficacy for fine-tuning outcomes within a constant model size remains uncertain. Practically, dependable predictors are essential to avoid the prohibitive expense of fine-tuning numerous checkpoints. This requirement is especially pronounced for monitoring and guiding decisions throughout the lengthy pre-training cycles (often months) of very large models (Liu et al., 2024a; Grattafiori et al., 2024), and also when subsequent fine-tuning involves substantial datasets, including potentially stopping unpromising runs early.

To investigate the predictability of fine-tuning outcome within feasible computational limits, our study employs a controlled methodology using smaller models. We train multiple variants of a 1B-parameter language model, each incorporating systematic variations in its pre-training configuration. We then evaluate the accuracy of potential predictors by comparing their values at the final pre-training checkpoints against the models' eventual performance after supervised fine-tuning (SFT). While

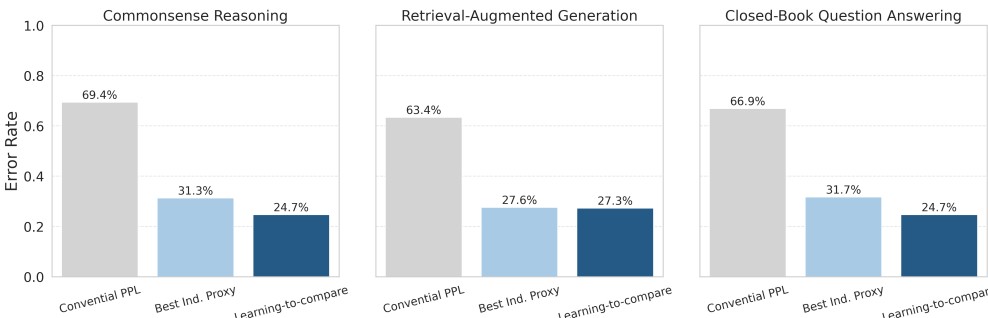

Figure 1: Mean pairwise error rates across three SFT tasks (separate plots). Each plot compares perplexity, the best individual proxy (Section 3), and the learning-to-compare proxy (shown on the x-axis). The y-axis represents the error rate, defined as the proportion of mis-classified LLM pairs regarding post-SFT performance.

simplified, we posit that this approach provides representative insights into the core question–whether fine-tuning outcome can be reliably predicted during and after pre-training. Specifically, we generated 50 distinct 1B-parameter LLM variants by systematically altering pre-training objectives (Raffel et al., 2020; Tay et al., 2023a;b), data composition strategies (Shen et al., 2024), and data processing techniques such as filtering and domain tagging (Penedo et al., 2024). These pre-trained models were subsequently fine-tuned across a diverse suite of tasks, including commonsense reasoning, retrieval-augmented generation and closed-book question answering. Specifically, we select five datasets (Clark et al., 2019; Zellers et al., 2019; Bisk et al., 2019; Mihaylov et al., 2018; Sakaguchi et al., 2021) for commonsense reasoning, four (Kwiatkowski et al., 2019; Joshi et al., 2017; Yang et al., 2018; Ho et al., 2020) for retrieval-augmented generation, and two (Kwiatkowski et al., 2019; Joshi et al., 2017) for closed-book question answering. To align with the practical model development scenarios where the primary goal is to identify top performers from a set of candidate models, we formulate the prediction challenge as a pairwise classification task: given two pre-trained models differing only in pre-training, the goal is to predict which model will achieve superior performance after SFT.

We begin by evaluating conventional perplexity, computed using a causal language modeling objective (Brown et al., 2020), as a predictor of SFT performance. Surprisingly, this standard metric correlates poorly with the downstream results of the LLMs after fine-tuning, resulting in prediction error rates exceeding 60% across all three evaluated tasks–worse than the 50% error rate of random guessing (Figure 1). Motivated by prior work (Raffel et al., 2020; Tay et al., 2023a; Von Oswald et al., 2023), we then introduce alternative pre-training available proxies, including span corruption-based perplexity and k-shot learning performance (Min et al., 2022). These proxies yield substantially improved prediction accuracy; the best-performing proxy for each task reduces the error rate by nearly half compared to conventional perplexity (Figure 1). For example, in the commonsense reasoning task, the error rate drops from 69.4% to 31.3%. Furthermore, we propose a learning-to-compare (LTC) framework that integrates multiple proxies via supervised classification. By learning inter-actions across these heterogeneous signals, the LTC approach achieves more robust performance estimation and further decreases the predictive error. The contributions of this paper are three-folds.

- We present the first formal study focused on predicting post-SFT performance across LLMs of identical size using pretraining signals—departing from prior scaling-based analyses.

- Our work demonstrates the insufficiency of perplexity for this prediction task and introduces novel unsupervised and supervised proxies achieving over a 50% reduction in error rates.

- Our work underscores the challenges of predicting supervised fine-tuning performance and confirms the practical value of the proposed proxies in specific scenarios; to foster further research, we provide the SFT performance data and individual pre-training proxy measurements in Appendix Table 8.

## 2 PROBLEM DEFINITION AND SETUP

This section defines the problem and details the setup, including the generation of diverse LLM variants, the target SFT tasks, and the pre-training signals used as prediction proxies.

### 2.1 LLM VARIANTS AND TARGET SFT TASKS

**LLM model variations.** To approximate pre-training studies while maintaining reasonable computational resources, we continuously trained a 1B parameter LLM with 100B tokens, systematically ablating pre-training objectives, data mixture re-weighting, and data filtering and tagging. This continuous pre-training approach allowed us to generate a wider range of model variants while managing computational resources. **Pre-training objectives:** We explored seven pre-training objectives: causal language modeling (CLM) (Brown et al., 2020), span corruption (SC) (Raffel et al., 2020), prefix language modeling (PLM) (Raffel et al., 2020), SC+CLM, UL2 (Tay et al., 2023a), UL2R (Tay et al., 2023b), and UL2R+CLM (Garcia et al., 2023). CLM and PLM generate tokens left-to-right, with CLM using the full context and PLM conditioning on a prefix. SC reconstructs masked spans, parameterized by noise density and mean span length, set to $(0.15, 3)$ following (Raffel et al., 2020). SC+CLM jointly trains SC and CLM. UL2 mixes six SC variants with PLM, while UL2R uses two SC settings—$(0.15, 3)$ and $(0.5, 32)$—with PLM. UL2R+CLM extends UL2R by adding a CLM objective. **Mixture re-weighting:** We train on the 627B-token Slimpajama corpus (Soboleva et al., 2023), which includes seven diverse domains. We reweigh different domains following (Shen et al., 2024), producing six 100B-token subsets by adjusting domain distributions (detailed in Table 5 in Appendix); **Data filtering and tagging:** Source domain metadata was integrated by pre-pending each instance with its respective domain label (e.g., [Common Crawl]). Length-based sub-corpora were generated by selecting instances within the [25%, 75%] and [75%, 100%] token length quantiles. We in total produced 50 distinct LLM variants, the specifications of which are provided in Table 6 in Appendix.

**Target SFT tasks.** We employed commonsense reasoning (CMS), retrieval-augmented generation (RAG), and closed-book question answering (CBQA) as the target supervised fine-tuning (SFT) tasks. These tasks were chosen to assess critical LLM capabilities such as reasoning, context utilization, and memorization, which are complex and challenging. Furthermore, they are well-established within the NLP community and offer ample training data. To obtain task-level SFT scores, we averaged dataset-specific scores within each task. Specifically, CMS included BoolQ (Clark et al., 2019), PIQA (Bisk et al., 2019), HellaSwag (Zellers et al., 2019), Winogrande (Sakaguchi et al., 2021), and OpenBookQA (Mihaylov et al., 2018); RAG utilized NQ (Kwiatkowski et al., 2019), TriviaQA (Joshi et al., 2017), HotpotQA (Yang et al., 2018), and 2Wiki (Ho et al., 2020); and CBQA used NQ (Kwiatkowski et al., 2019) and TriviaQA (Joshi et al., 2017).

### 2.2 PREDICTION PROXIES

This study investigates two distinct prediction proxies: Perplexity (PPL) and k-shot learning (Kshot). Perplexity is a prevalent prediction proxy for monitoring LLM pre-training, whereas the intuitive rationale for k-shot learning lies in its potential correlation with fine-tuned performance on the identical task (Tay et al., 2023a; Ahn et al., 2023; Von Oswald et al., 2023).

Perplexity (PPL) is calculated through two distinct methods. PPL-CLM represents the conventional causal language modeling perplexity. Driven by UL2's (Tay et al., 2023a) demonstration of span corruption's efficacy in supervised fine-tuning, we present the PPL-SC proxy. This metric is derived from the span corruption methodology, as in T5 (Raffel et al., 2020), and computes perplexity over randomly sampled text spans. Both perplexities are computed on the PILE development set (Gao et al., 2020), with span corruption parameters $(0.15, 3)$ (Raffel et al., 2020). For the purposes of clarity in presentation, we utilize the inverse of the actual perplexity values, namely, $\frac{1}{\text{Perplexity}}$. This transformation aligns with Kshot such that higher proxy values correspond to improved SFT performance. Unless explicitly stated otherwise, *PPL-CLM* and *PPL-SC* in this paper refer to these inverted values. K-shot performance is calculated by averaging the results from evaluating test sets of target datasets for each SFT task. The actual prompts are detailed in Appendix F. Akin to Chowdhery et al. (2023), we use 1 shot for CMS and 5 shots for RAG and CBQA. This yields five efficient proxy scores for each model: *PPL-CLM*, *PPL-SC*, *Kshot-CMS*, *Kshot-RAG*, and *Kshot-CBQA*.

|  | SFT-CMS | SFT-RAG | SFT-CBQA |
|---|---|---|---|
| **Conventional Perplexity** | | | |
| PPL-CLM | .332 | .380 | .354 |
| **Individual Prediction Proxies** | | | |
| PPL-SC | .703 | .622 | .609 |
| Kshot-CMS | .573 | .569 | .525 |
| Kshot-RAG | .696 | **.766** | **.704** |
| Kshot-CBQA | .437 | .447 | .467 |
| **Aggregated Prediction Proxies** | | | |
| Combine Five Proxies | .622 | .598 | .564 |
| **Analytical Exploration of Headroom Potential** | | | |
| PPL-SC + Kshot-RAG | .744 | .696 | .642 |
| PPL-SC + Kshot-RAG - PPL-CLM | .763 | .692 | .635 |

Table 1: Accuracy of Individual vs. Aggregated Proxy Predictors.

## 2.3 PAIRWISE ACCURACY AS A MEASURE OF PREDICTIVE POWER

We evaluated each pre-trained LLM variant by fine-tuning it on individual target dataset training sets and assessing performance on the corresponding evaluation sets. Task-level scores (SFT-CMS, SFT-RAG, SFT-CBQA) were computed by averaging these dataset results. Since practical model selection often involves choosing the best from a small candidate pool, our primary analysis focused on evaluating the discriminating power of prediction proxies (like perplexity). To achieve this, we formulated the evaluation as a pairwise prediction task. We generated all 1225 unique pairs from the 50 LLM variants and measured how accurately each proxy could predict which model in a pair would achieve better aggregated task-level SFT performance. This pairwise prediction accuracy is our main metric for proxy effectiveness.

## 3 PREDICTIVE POWER ON SFT TASKS

**Accuracy of individual prediction proxies to SFT performance.** Table 1 details the pairwise SFT prediction accuracy of various proxy metrics across 50 LLM variants. Conventional perplexity (PPL-CLM) exhibited low accuracy (e.g., 0.3 on SFT-CMS), contrasting sharply with its known correlation strength in scaling studies. The span corruption perplexity (PPL-SC) performed better ($>$ 0.5 accuracy), consistent with prior findings on span corruption benefits (UL2) (Tay et al., 2023a). Few-shot (k-shot) proxies achieved higher accuracy still, with Kshot-RAG reaching $\approx 0.7$ on SFT-CMS and SFT-RAG. Despite these improvements, no single proxy proved universally reliable across all tested SFT tasks.

**Aggregating diverse prediction proxies.** We explore improving prediction by combining normalized proxy scores (details in Table 1). While averaging all five proxies underperforme Kshot-RAG alone, combining PPL-SC and Kshot-RAG matched Kshot-RAG's performance and surpass PPL-SC. Despite these improvements, even the best individual or combined proxies yield pairwise error rates around 30%, suggesting inherent task difficulty limits performance. Nevertheless, these simple arithmetic combinations (e.g., PPL-SC + Kshot-RAG - PPL-CLM) demonstrate the potential to outperform individual proxies through effective aggregation.

**A predictive power case study using varied pre-training objectives.** To understand proxy limitations, we analyzed how well PPL-CLM, PPL-SC, and Kshot-RAG predict relative SFT performance between models differing only in their pre-training objective. We grouped models by objective (CLM, SC, UL2, etc.) and evaluated pairwise prediction accuracy for comparisons between these groups (details in Figure 2; Appendix C covers data variations). Confirming earlier results, PPL-SC and Kshot-RAG consistently outperformed PPL-CLM. However, their accuracy depended significantly on two factors: (1) The specific pre-training difference: Proxies better captured large performance gaps caused by different objectives (e.g., SC vs. CLM, often $\geq 0.6$ accuracy) than smaller variations.

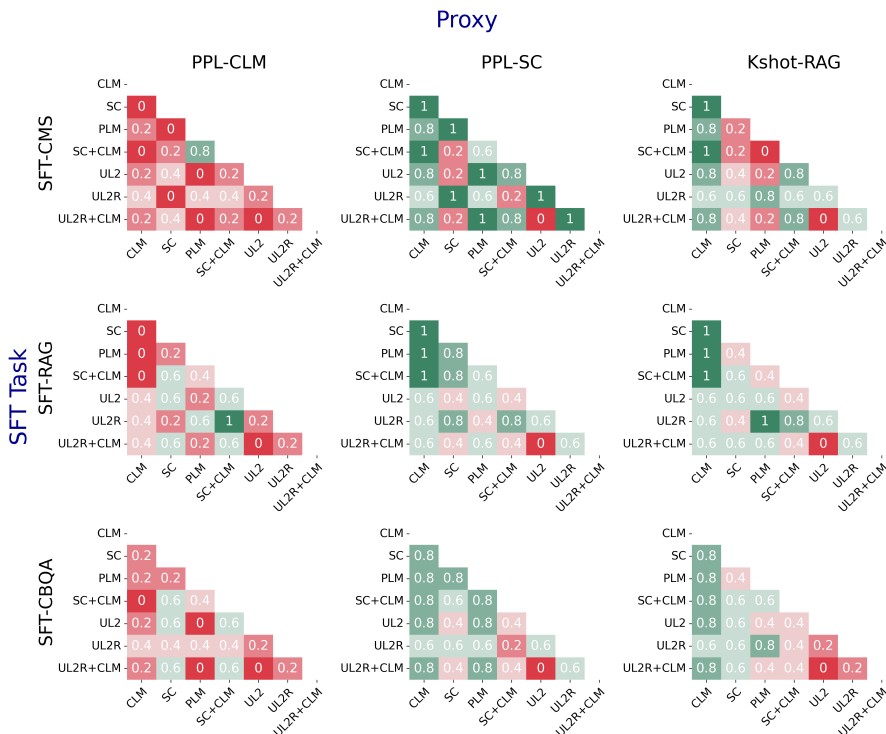

Figure 2: Pairwise prediction accuracy for PPL-CLM, PPL-SC, and Kshot-RAG comparing LLMs differing only in pre-training objective, across three SFT tasks (rows) and the three proxies (columns). Each cell indicates average accuracy of pairs where the proxy prediction agreed with the SFT result.

(2) The target SFT task: A specific comparison (e.g., SC vs. SC+CLM) could yield low accuracy on one task (SFT-CMS, 0.2) but high accuracy on others (SFT-RAG/SFT-CBQA, $\geq 0.6$).

## 4    LEARNING TO COMPARE

Recognizing the complementary strengths of individual proxies amidst their challenges (Section 3, Table 1, Figure 2), we now explore supervised classifiers to combine these signals for potentially enhanced SFT performance prediction.

### 4.1    FORMULATION

Given two LLMs $m_i$ and $m_j$, our goal is to predict which model achieves better downstream SFT performance. We denote the values of the five proxies for each model $m_i$ as $\{P_{m_i}^k\}_{k \in \mathcal{D}}$, where $\mathcal{D} = \{\text{PPL-CLM}, \text{PPL-SC}, \text{Kshot-CMS}, \text{Kshot-RAG}, \text{Kshot-CBQA}\}$. The learning-to-compare model leverages these proxies by training a binary classifier $f$ to predict the fine-tuned performance comparison between model pair $(m_i, m_j)$. For each proxy $k$, we construct the feature vector: $h_k(p_{m_i}, p_{m_j}) = \left[ p_{m_i}^k - p_{m_j}^k, \; p_{m_i}^k \cdot p_{m_j}^k, \; p_{m_i}^k, \; p_{m_j}^k \right] \in \mathbb{R}^4$. We concatenate features from all five proxies to form the input and lead to 20 features, namely, $H(p_{m_i}, p_{m_j}) \in \mathbb{R}^{20}$. We define the ground-truth label $y_{ij}$ as a binary value, where $y_{ij} = 1$ if LLM $m_i$ performs better after SFT than $m_j$, and $y_{ij} = 0$ otherwise. The classifier is trained by minimizing the binary cross-entropy loss (formulation is provided in Appendix Section D).

### 4.2    EXPERIMENT SETUP

We implemented the supervised classifier using LightGBM (details on other models in Appendix Section D), training separate models per SFT task (CMS, RAG, CBQA). To ensure robustness,

| | SFT-CMS | SFT-RAG | SFT-CBQA |
|---|---|---|---|
| **Conventional Perplexity** | | | |
| PPL-CLM | $.306\pm_{.081}$ | $.366\pm_{.060}$ | $.331\pm_{.054}$ |
| **Individual and Aggregated Proxies** | | | |
| Kshot-RAG | $.687\pm_{.073}$ | $.724\pm_{.047}$ | $.683\pm_{.077}$ |
| Combine Five Proxies | $.612\pm_{.055}$ | $.585\pm_{.051}$ | $.540\pm_{.104}$ |
| **Learning To Compare (% Relative to Kshot-RAG)** | | | |
| **Trained on the target task** | | | |
| Learning-to-compare | $\mathbf{.753}\pm_{.054}$ (+9.6%) | $\mathbf{.727}\pm_{.039}$ (+0.4%) | $\mathbf{.753}\pm_{\mathbf{.060}}$ (+10.2%) |
| **Trained on the source task** | | | |
| SFT-CMS (Src) | $.753\pm_{.054}$ (+9.6%) | $.712\pm_{.054}$ (-1.7%) | $.707\pm_{.057}$ (+3.3%) |
| SFT-RAG (Src) | $.734\pm_{.047}$ (+6.8%) | $.727\pm_{.039}$ (+0.4%) | $.717\pm_{.071}$ (+5.0%) |
| SFT-CBQA (Src) | $.734\pm_{.052}$ (+6.8%) | $.718\pm_{.050}$ (-0.1%) | $.753\pm_{.060}$ (+10.2%) |

Table 2: Pairwise prediction accuracy (mean $\pm$ std dev, 20 runs): Unsupervised baselines vs. supervised classifiers on SFT-CMS, SFT-RAG, SFT-CBQA.

we performed 20 runs, each using a random 60%/40% split of the 50 LLM variants to generate training/testing pairs (splits varied per run). We report mean accuracy and standard deviation over the 20 runs in Table 2 (middle section), compared against unsupervised baselines including PPL-CLM and Kshot-RAG.

### 4.3 RESULTS

**Learning-to-compare enhances predictive power beyond the best-performing proxies.** Despite the challenges of constructing prediction proxies, supervised learning significantly enhances predictive performance compared to individual or aggregated proxies. As shown in Table 2, LightGBM outperforms the best individual proxy, Kshot-RAG, by a substantial margin on the SFT-CMS and SFT-CBQA tasks, improving predictive power by 10% while maintaining comparable performance on SFT-RAG. This confirms that combining diverse proxies can further boost predictive accuracy.

**Learning-to-compare generalizes well across different target tasks.** We further assessed Light-GBM's generalization by training on one SFT task (source) and evaluating on others (target), using all five proxies as input. The aim was to determine if a classifier learned for one task could predict performance on different ones. Results (Table 2, bottom section) reveal effective generalization: models trained on a source task maintained high predictive accuracy on target tasks, typically performing within 2-3% of classifiers trained directly on the target task. This demonstrates the robustness of the learning-to-compare approach across different SFT domains without significant performance loss.

**Proxy importance.** We quantify each proxy's contribution to the LightGBM classifiers by computing their normalized gain-based importance scores, as illustrated in Figure 3 (detailed in Appendix Section E). Kshot-RAG consistently emerged as the most influential proxy across the three SFT tasks, showing particular dominance in SFT-RAG and SFT-CBQA. PPL-SC and PPL-CLM represented the next tier of importance; for instance, PPL-SC was second most important for SFT-CMS, while PPL-CLM ranked second for SFT-CBQA. Intriguingly, PPL-CLM contributed more significantly to the LightGBM model's predictions than Kshot-CMS and Kshot-CBQA, despite possessing lower standalone accuracy (Table 1). Our hypothesis is that the supervised classifier effectively utilizes the strong negative correlation observed between PPL-CLM and SFT task performance.

## 5 CAN POST SFT LLM PERFORMANCE BE RELIABLY PREDICTED?

While the learning-to-compare method doubles prediction accuracy over perplexity (Table 2), its persistent 25% pairwise error rate limits general applicability. This section analyzes its practical utility. Analysis shows pairwise prediction accuracy depends heavily on the magnitude of the actual

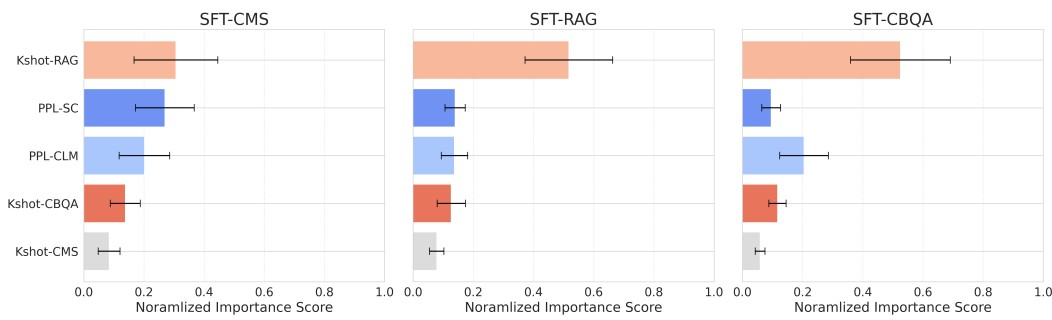

Figure 3: Relative influence of proxy metrics in the LTC framework (LightGBM).

SFT performance difference, proving less reliable for subtle distinctions. But we demonstrate reliable recall of top models within small candidate sets, suggesting value for initial model filtering.

## 5.1 IMPACT OF PERFORMANCE GAPS ON PREDICTION RELIABILITY

Predicting the relative performance between two language models is expected to be more reliable when their actual performance levels are significantly different. Conversely, distinguishing between models with similar performances poses a greater challenge. This section investigates how the magnitude of the performance gap between model pairs influences the reliability of our prediction classifiers.

To explore the relationship between performance disparity and classifier accuracy, we first calculated the absolute difference in supervised fine-tuning (SFT) performance for each model pair on the target task. We hypothesized that classification accuracy would correlate positively with the size of this performance gap. For quantitative analysis, we categorized the model pairs into five quantiles based on their true post-SFT performance difference: [0–20%], [20–40%], [40–60%], [60–80%], and [80–100%]. Subsequently, we evaluated and compared the classification accuracy for three predictors—PPL-CLM, Kshot-RAG, and Learning-to-compare—within each quantile. These results are visualized in Figure 4.

The findings show that prediction reliability for both Kshot-RAG and the Learning-to-compare predictors indeed improves as the performance gap between models widens. For pairs with minimal performance differences ([0–20%] quantile), where models perform almost identically after fine-tuning, prediction accuracy is low, near chance levels (approximately 0.5). As the absolute performance difference increases, accuracy steadily rises, reaching approximately 0.9 for the most distinct pairs ([80–100%] quantile). This confirms that these classifiers yield more reliable predictions when comparing models that are easier to distinguish. Interestingly, PPL-CLM demonstrates the opposite behavior: its accuracy diminishes as the performance gap increases, further highlighting that conventional perplexity is not a dependable indicator for this prediction scenario. Among the methods tested, the learning-to-compare classifier consistently outperformed both PPL-CLM and Kshot-RAG across the quantiles, showing particular strength on the SFT-CMS and SFT-CBQA tasks.

## 5.2 RECALL THE BEST MODEL FROM A SMALL CANDIDATE SET

One key practical use for LLM performance predictors is to identify the most promising models within a group of candidates, which can lead to significant cost savings by reducing the number of models that undergo supervised fine-tuning. To assess our classifier's effectiveness in this critical application—specifically, its ability to recall the best pre-trained LLMs—we performed a ranking experiment where pairwise comparisons between models were predicted and then aggregated into an overall ranking using Borda Count scoring Dwork et al. (2001). Specifically, for each model $m_i$, we compute its total score by counting the number of pairwise wins over all other models.

$$\text{Score}(m_i) = \sum_{j \neq i} \mathbf{1}_{(f(m_i, m_j) > 0.5)}$$

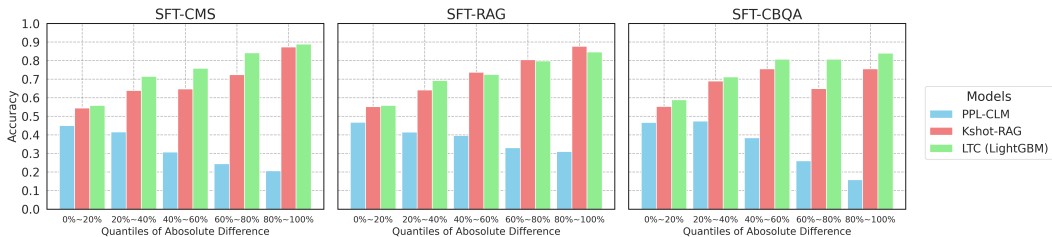

Figure 4: Accuracy comparison of PPL-CLM, Kshot-RAG, and Learning-to-Compare (LTC) on SFT tasks (CMS, RAG, CBQA), grouped into five quantiles by absolute SFT performance difference.

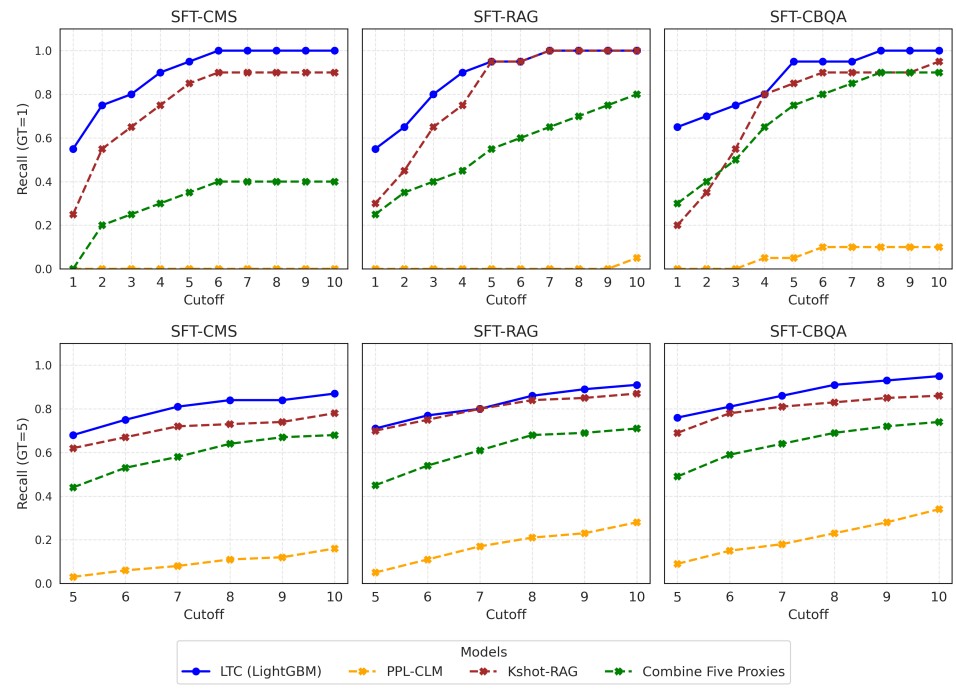

Figure 5: Top-1 (top row) and Top-5 (bottom row) recall comparison at various cutoffs: supervised Learning-to-compare (LTC) vs. unsupervised baselines on SFT-CMS, SFT-RAG, and SFT-CBQA tasks.

where $f(m_i, m_j)$ denotes the classifier's predicted probability that $m_i$ outperforms $m_j$. $\mathbf{1}_{(\cdot)}$ is the indicator function. Finally, models are ranked based on their total scores, with higher scores indicating better predicted fine-tuned performance. Models achieving more pairwise 'wins' received higher scores, indicating better predicted performance. The evaluation results, presented as top-1 and top-5 recall in Figure 5, show that our "learning-to-compare" method consistently identified the top-performing LLMs. Impressively, it achieved perfect top-1 recall for the SFT-CMS, SFT-RAG, and SFT-CBQA tasks by focusing on the top 7, 7, and 8 predicted models respectively, demonstrating its effectiveness even when narrowing down a relatively small candidate pool (as few as 8 models). Additionally, the unsupervised Kshot-RAG method showed strong performance, corroborating observations from Section 3.

# 6 RELATED WORK

**Pre-training of LLMs** LLM pre-training fundamentally shapes capabilities like reasoning (Wei et al., 2022; Kojima et al., 2022; Zellers et al., 2019), knowledge (Chang et al., 2024), and tool

use (Yao et al., 2023; Mo et al., 2023). Critical pre-training design choices include the training objective—such as dominant CLM (Brown et al., 2020; OpenAI, 2023) for generation, SC (Raffel et al., 2020) which aids fine-tuning (Tay et al., 2023a), or combined UL2-style approaches (Tay et al., 2023a;b; Garcia et al., 2023), potentially using PrefixLM (Du et al., 2022; Chowdhery et al., 2023)—and pre-trained corpus composition, which involves quality curation (Rae et al., 2021; Touvron et al., 2023), filtering (Penedo et al., 2023; Xia et al., 2024), and source mixing (Weber et al., 2024; Shen et al., 2024) to ensure broad coverage and robustness. Given the variety of design options, lightweight methods to predict final performance are highly desirable for efficient model development. This work investigates predictors for supervised fine-tuning outcomes, utilizing systematic variations across several pre-training design factors in our study.

**LLMs SFT Performance Prediction**    The ability to predict the performance of large language models (LLMs) after fine-tuning has gained significant importance, largely driven by the substantial computational investment required for pre-training. Previous research (Kaplan et al., 2020; Hoffmann et al., 2022; Henighan et al., 2020) established scaling laws showing that increasing pre-training FLOPs typically reduces perplexity on held-out data, correlating with enhancements in capabilities like chain-of-thought reasoning (Wei et al., 2022; Kojima et al., 2022), preference alignment (Ouyang et al., 2022; Bai et al., 2022), and multilingual understanding (Chowdhery et al., 2023), suggesting larger models generally yield better downstream performance. Analogous scaling phenomena, where lower perplexity often corresponds to improved outcomes, have also been noted when fine-tuning LLMs for specific applications (Zhang et al., 2024; Isik et al., 2025); for instance, Isik et al. (2025) reported such a correlation for machine translation performance. However, token-level perplexity can over-weight frequent tokens and mask deficits on rare or semantically critical ones (Sinha & et al., 2020), and its predictability has been questioned for long-context generation (Liu et al., 2024b; Fang et al., 2025) and many-shot in-context learning (Agarwal et al., 2024), implying it may not be a robust indicator across all downstream tasks.

Departing from scaling-law studies across varying model sizes or from settings focused on extreme input/output lengths, we evaluate the efficacy of perplexity as a predictor of fine-tuned performance among same-size LLMs trained with the same pre-training compute, on widely used NLP tasks. In this controlled regime, we find that perplexity is not a reliable predictor of downstream SFT performance, calling into question its utility as a one-size-fits-all proxy for selecting among equal-size LLM variants. Building on this observation, we introduce several pre-training accessible proxies that exhibit stronger correlations with downstream SFT outcomes. And further propose a learning-to-compare framework that ensembles these proxies to rank candidate models, yielding consistent gains over any single proxy and outperforming perplexity-based selection.

## 7    CONCLUSION AND FUTURE DIRECTIONS

This study focused on the challenge of predicting LLM performance after supervised fine-tuning (SFT) using only pre-training indicators, establishing that conventional perplexity is unreliable for this purpose. We approached this as a pairwise classification task, using 1B parameter LLM variants with diverse pre-training configurations. We introduced novel unsupervised (Kshot-RAG, PPL-SC) and supervised ("learning-to-compare") proxy metrics, which successfully reduced relative performance prediction error by over 50% compared to perplexity. These proxies proved effective for predicting outcomes, particularly between models with large performance gaps, and for identifying top-performing candidates, thereby enabling more efficient LLM development pathways.

Future work should explore the generalizability of these findings to larger model scales and a broader range of downstream tasks and fine-tuning paradigms. Further investigation into a broader array of pre-training strategies, data compositions, and the development of even more sophisticated proxy metrics could yield deeper insights. Additionally, exploring the theoretical connections between specific pre-training objectives or data characteristics and their influence on downstream task adaptability after fine-tuning represents a promising avenue for future work, ultimately enabling more efficient LLM development and selection.

# 8 ADDITIONAL EXPERIMENTS AND CLARIFICATIONS (REBUTTAL UPDATES)

In this section, we provide additional experimental results in response to reviewer feedback, covering extended perplexity evaluations, benchmark resolution analysis, and feature synergy ablation studies.

## 8.1 EXTENDED PERPLEXITY ANALYSIS (SLIMPAJAMA SPLITS & BENCHMARK TRAIN SETS)

To address the concern that perplexity on the Pile dev set might be insufficient, we conducted a comprehensive evaluation of additional perplexity-based proxies. We computed perplexity on:

- **SlimPajama-Wiki**: Domain-specific perplexity on the Wikipedia split.

- **SlimPajama-CC**: Domain-specific perplexity on the CommonCrawl split.

- **Benchmarks PPL-CLM**: Mean perplexity computed directly on the training sets of the downstream benchmarks.

We evaluated these proxies using pairwise classification accuracy (Higher is Better). As shown in Table 3, all perplexity-based variants, even those matched to the domain, yield significantly lower predictive accuracy (mean accuracy $< 0.46$) compared to Kshot-RAG (mean accuracy $0.72$). This confirms our central claim that perplexity is not a reliable indicator of relative post-SFT performance in this setting.

Table 3: Pairwise predictive accuracy (Higher is Better) of extended perplexity-based proxies compared to Kshot-RAG.

| Proxy Indicator | SFT-CBQA | SFT-CMS | SFT-RAG | Mean Acc. |
|---|---|---|---|---|
| **Kshot-RAG** (Ours) | **0.7064** | **0.6972** | **0.7685** | **0.7240** |
| Benchmarks PPL-CLM | 0.4607 | 0.4133 | 0.4863 | 0.4534 |
| SlimPajama-Wiki PPL | 0.4450 | 0.3813 | 0.4664 | 0.4309 |
| PPL-CLM (Pile-dev) | 0.3350 | 0.3101 | 0.3635 | 0.3362 |
| SlimPajama-CC PPL | 0.3110 | 0.2881 | 0.3212 | 0.3067 |

## 8.2 BENCHMARK RESOLUTION ANALYSIS (TOP-4 BENCHMARKS)

To verify that the advantage of Kshot-RAG is not due to noise in low-resolution benchmarks, we selected the top-4 benchmarks with the highest resolution: CBQA_TriviaQA, CBQA_NQ, Winogrande, and OpenBookQA. We computed the mean pairwise **accuracy** (Higher is Better) for each proxy on this high-resolution subset.

As detailed in Table 4, Kshot-RAG achieves the highest mean accuracy (0.6963), while Pile-CLM (perplexity) shows the lowest accuracy (0.3649). This consistent trend demonstrates that the predictive advantage of Kshot-RAG holds robustly across high-resolution tasks and is not an artifact of benchmark granularity.

Table 4: Mean pairwise **accuracy** (Higher is Better) on the Top-4 high-resolution benchmarks. Kshot-RAG consistently achieves the highest accuracy.

| Metric | TriviaQA | NQ | Winogrande | OpenBookQA | Mean Acc. |
|---|---|---|---|---|---|
| **Kshot-RAG** | **0.7192** | **0.6841** | **0.6947** | **0.6873** | **0.6963** |
| Pile-SC | 0.6024 | 0.5494 | 0.6424 | 0.6735 | 0.6169 |
| Kshot-CMS | 0.5159 | 0.5910 | 0.5706 | 0.6016 | 0.5698 |
| KShot-CBQA | 0.4645 | 0.5371 | 0.4661 | 0.4980 | 0.4914 |
| Pile-CLM | 0.3404 | 0.4555 | 0.2931 | 0.3706 | 0.3649 |

## 8.3 FEATURE SYNERGY AND ABLATION STUDY

To clarify the mechanism behind proxy combination, we conducted an exhaustive ablation study using a LightGBM classifier across 20 random seeds, enumerating all possible feature subsets from size k=1 to k=5. For each subset size k, we selected the combination achieving the highest accuracy and visualized these results in Figure 6.

**Observations:** As illustrated in Figure 6, at the $k = 1$ baseline, **Pile-SC** is the most effective predictor for SFT-CMS (Acc $\approx 0.61$), while **Kshot-RAG** dominates for SFT-RAG and SFT-CBQA (Acc $\approx 0.58$ and $0.61$). Crucially, combining just two features ($k = 2$) yields substantial performance gains: c

- **SFT-CMS**: +11.67% improvement.
- **SFT-RAG**: +12.06% improvement.
- **SFT-CBQA**: +9.49% improvement.

**Conclusion:** These findings provide strong evidence of feature synergy: the model combines complementary signals to achieve substantially higher accuracy than what any single feature can provide. We also observe a consistent upward trend (except $k=4$ to 5 in SFT-RAG). in performance as more features are included

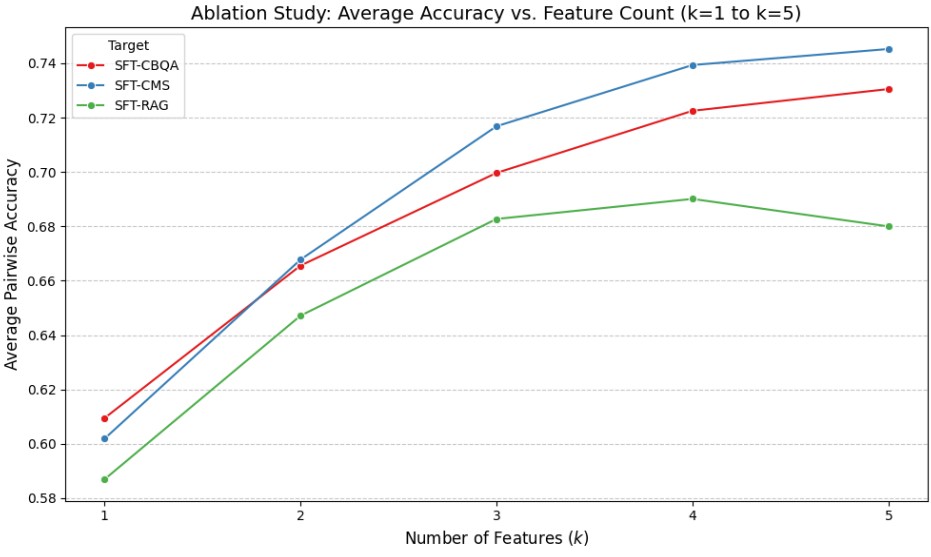

Figure 6: Ablation study showing average pairwise accuracy vs. number of features ($k$).

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

## A  LLM USAGE STATEMENT

We employed large language models (LLMs) mainly for refining the authors' original writing, aiming to improve clarity and readability.

| | Sub Dataset | DC-0 | DC-1 | DC-2 | DC-3 | DC-4 | DC-5 |
|---|---|---|---|---|---|---|---|
| | Commoncrawl | 52.2% | 100.0% | 90.9% | 75.8% | 75.8% | 75.8% |
| | C4 | 26.7% | 0.0% | 0.0% | 0.0% | 0.0% | 0.0% |
| | GitHub | 5.2% | 0.0% | 9.1% | 24.2% | 0.0% | 9.1% |
| SlimPajama | Books | 4.2% | 0.0% | 0.0% | 0.0% | 0.0% | 7.9% |
| | ArXiv | 4.6% | 0.0% | 0.0% | 0.0% | 0.0% | 0.0% |
| | Wikipedia | 3.8% | 0.0% | 0.0% | 0.0% | 24.2% | 7.3% |
| | StackExchange | 3.3% | 0.0% | 0.0% | 0.0% | 0.0% | 0.0% |

Table 5: six configurations of sub dataset combinations in Slimpajama

## B   PRETRAINING AND LLMS

We use SlimPajama Soboleva et al. (2023) as our pretraining corpus, which consists of data from seven domains. Following Shen et al. (2024), we apply domain re-weighting to create six dataset variants. The detailed domain proportions for each variant are provided in Table 5.

We pretrain 50 LLMs, each with 1 billion parameters, on 100 billion tokens. Model variants are generated by varying pretraining objectives, dataset composition strategies, and learning rates. The detailed pretraining configuration for each model is provided in Table 6.

## C   PROXY PREDICTIVE ACCURACY

Similar to Section 3, we group the pre-trained LLMs into six categories either based on their domain re-weighting or tagging & length filtering configurations. In both cases, paired models share the same pretraining configurations except for the group-specific factor (domain re-weighting or tagging & length filtering). We compute the predictive accuracy of each proxy on three SFT tasks and report the results in the Figure 7 and Figure 8.

## D   CLASSIFIER IMPLEMENTATION DETAIL

**Loss function**: Assuming the LLMs in training set as $\mathcal{M}_{train}$, we train the classifier using the binary cross-entropy loss.

$$\mathcal{L} = \frac{1}{C} \sum_{m_i, m_j \in \mathcal{M}_{train} \text{ and } i \neq j} -y_{ij} \log f\left(H(p_{m_i}, p_{m_j})\right) - (1 - y_{ij}) \log\left(1 - f\left(H(p_{m_i}, p_{m_j})\right)\right)$$

Where $C$ is the total number of pairs in $\mathcal{M}_{train}$ equals to $\frac{|\mathcal{M}_{train}|(|\mathcal{M}_{train}|-1)}{2}$.

We also instantiate the learning-to-compare framework using Logistic Regression and Neural Networks as backbone models. Their performance, compared with unsupervised baselines, is reported in Table 7.

The implementation details are as follows: For logistic regression, we use scikit-learn's (Pedregosa et al., 2011) LogisticRegression with the default lbfgs solver for binary classification. The model applies $L_2$ regularization with strength $C = 1.0$, fits an intercept, and runs up to 100 iterations. Class weighting is not applied. For the neural network, we use scikit-learn's MLPClassifier with two hidden layers of size 32 each and ReLU activation. The model is optimized using the Adam solver and trained for a maximum of 100 iterations. All other hyperparameters are set to their default values. For LightGBM, we use the LGBMClassifie from the official lightgbm library [1]. The objective is set to binary with binary_logloss as the evaluation metric. All other hyperparameters follow the default settings: num_leaves=31, learning_rate=0.1, n_estimators=100, feature_fraction=1.0, bagging_fraction=1.0, and no regularization (lambda_l1=0.0, lambda_l2=0.0).

---

[1] https://lightgbm.readthedocs.io/en/latest/pythonapi/lightgbm.LGBMClassifier.html

| Model ID | Pretrained Objective | Domain Re-weight | LR | Domain Tagging | Length Filtering |
|---|---|---|---|---|---|
| 1 | CLM | DC-0 | 1e-4 | ✗ | ✗ |
| 2 | CLM | DC-0 | 2.5e-4 | ✗ | ✗ |
| 3 | CLM | DC-0 | 5e-4 | ✗ | ✗ |
| 4 | CLM | DC-0 | 7.5e-4 | ✗ | ✗ |
| 5 | CLM | DC-0 | 1e-3 | ✗ | ✗ |
| 6 | SC | DC-0 | 1e-4 | ✗ | ✗ |
| 7 | SC | DC-0 | 2.5e-4 | ✗ | ✗ |
| 8 | SC | DC-0 | 5e-4 | ✗ | ✗ |
| 9 | SC | DC-0 | 7.5e-4 | ✗ | ✗ |
| 10 | SC | DC-0 | 1e-3 | ✗ | ✗ |
| 11 | PLM | DC-0 | 1e-4 | ✗ | ✗ |
| 12 | PLM | DC-0 | 2.5e-4 | ✗ | ✗ |
| 13 | PLM | DC-0 | 5e-4 | ✗ | ✗ |
| 14 | PLM | DC-0 | 7.5e-4 | ✗ | ✗ |
| 15 | PLM | DC-0 | 1e-3 | ✗ | ✗ |
| 16 | SC+CLM | DC-0 | 1e-4 | ✗ | ✗ |
| 17 | SC+CLM | DC-0 | 2.5e-4 | ✗ | ✗ |
| 18 | SC+CLM | DC-0 | 5e-4 | ✗ | ✗ |
| 19 | SC+CLM | DC-0 | 7.5e-4 | ✗ | ✗ |
| 20 | SC+CLM | DC-0 | 1e-3 | ✗ | ✗ |
| 21 | UL2 | DC-0 | 1e-4 | ✗ | ✗ |
| 22 | UL2 | DC-0 | 2.5e-4 | ✗ | ✗ |
| 23 | UL2 | DC-0 | 5e-4 | ✗ | ✗ |
| 24 | UL2 | DC-0 | 7.5e-4 | ✗ | ✗ |
| 25 | UL2 | DC-0 | 1e-3 | ✗ | ✗ |
| 26 | UL2R | DC-0 | 1e-4 | ✗ | ✗ |
| 27 | UL2R | DC-0 | 2.5e-4 | ✗ | ✗ |
| 28 | UL2R | DC-0 | 5e-4 | ✗ | ✗ |
| 29 | UL2R | DC-0 | 7.5e-4 | ✗ | ✗ |
| 30 | UL2R | DC-0 | 1e-3 | ✗ | ✗ |
| 31 | UL2R+CLM | DC-0 | 1e-4 | ✗ | ✗ |
| 32 | UL2R+CLM | DC-0 | 2.5e-4 | ✗ | ✗ |
| 33 | UL2R+CLM | DC-0 | 5e-4 | ✗ | ✗ |
| 34 | UL2R+CLM | DC-0 | 7.5e-4 | ✗ | ✗ |
| 35 | UL2R+CLM | DC-0 | 1e-3 | ✗ | ✗ |
| 36 | CLM | DC-1 | 2.5e-4 | ✗ | ✗ |
| 37 | CLM | DC-2 | 2.5e-4 | ✗ | ✗ |
| 38 | CLM | DC-3 | 2.5e-4 | ✗ | ✗ |
| 39 | CLM | DC-4 | 2.5e-4 | ✗ | ✗ |
| 40 | CLM | DC-5 | 2.5e-4 | ✗ | ✗ |
| 41 | PLM | DC-1 | 2.5e-4 | ✗ | ✗ |
| 42 | PLM | DC-2 | 2.5e-4 | ✗ | ✗ |
| 43 | PLM | DC-3 | 2.5e-4 | ✗ | ✗ |
| 44 | PLM | DC-4 | 2.5e-4 | ✗ | ✗ |
| 45 | PLM | DC-5 | 2.5e-4 | ✗ | ✗ |
| 46 | CLM | DC-0 | 2.5e-4 | ✗ | [25% 75%] |
| 47 | CLM | DC-0 | 2.5e-4 | ✗ | [75% 100%] |
| 48 | CLM | DC-0 | 2.5e-4 | ✓ | ✗ |
| 49 | CLM | DC-0 | 2.5e-4 | ✓ | [25% 75%] |
| 50 | CLM | DC-0 | 2.5e-4 | ✓ | [75% 100%] |

Table 6: Pre-trained configurations of LLMs

# E  PROXY NORMALIZED IMPORTANCE SCORE FOR LIGHTGBM

We use LightGBM's gain-based feature importance, which quantifies how much each feature contributes to reducing the model's loss. Specifically, for each feature $f$, the importance is defined as the total reduction in the loss function (binary log-loss in our case) due to splits on that feature across all trees in the ensemble.

Let $\mathcal{T}$ denote the set of all decision trees in the trained LightGBM model. For each tree $t \in \mathcal{T}$ and each split node $s \in t$, let $f_s$ be the feature used at split $s$, and let $\Delta\mathcal{L}(s)$ denote the reduction in the

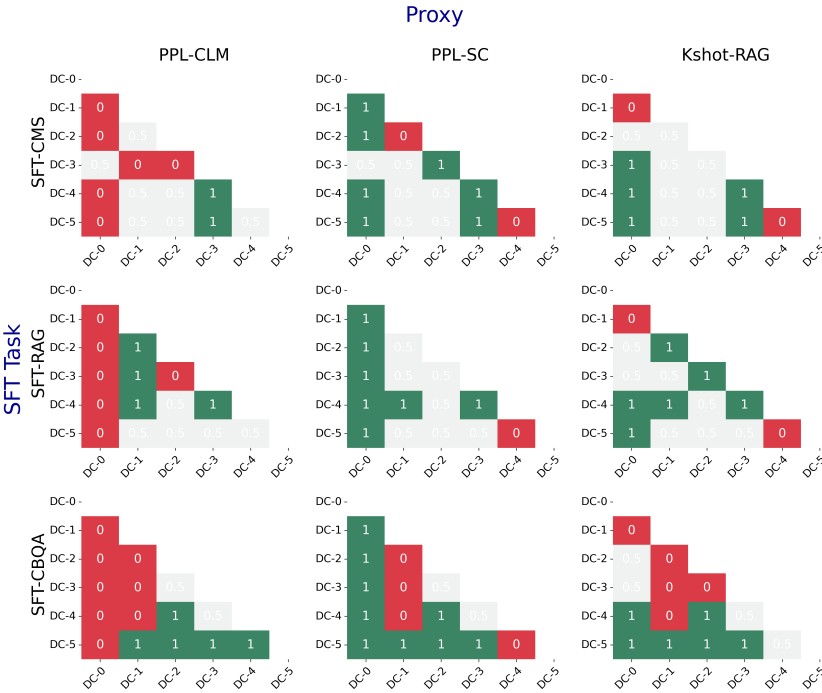

Figure 7: Predictive accuracy of PPL-CLM, PPL-SC, and Kshot-RAG in distinguishing the better-performing model between two LLMs with different pre-trained dataset domain re-weighting (other pre-trained configurations fixed). DC-0 to DC-5 referes to different dataset variants, detailed in Table 5.

loss function caused by that split. Then, the gain-based importance for feature $f$ is computed as:

$$\text{Gain}(f) = \sum_{t \in \mathcal{T}} \sum_{\substack{s \in t \\ f_s = f}} \Delta \mathcal{L}(s)$$

In our setting, we construct a 20-dimensional feature vector $H(p_{m_i}, p_{m_j}) \in \mathbb{R}^{20}$ for each model pair $(m_i, m_j)$ using five proxies, with each proxy contributing four dimensions as defined in:

$$h_k(p_{m_i}, p_{m_j}) = \left[ p^k_{m_i} - p^k_{m_j}, \ p^k_{m_i} \cdot p^k_{m_j}, \ p^k_{m_i}, \ p^k_{m_j} \right]$$

To compute proxy-level importance, we group every four dimensions corresponding to each proxy and sum their individual gain scores:

$$\text{Gain}(k) = \sum_{f \in \mathcal{F}_k} \text{Gain}(f)$$

where $\mathcal{F}_k$ denotes the set of four features derived from proxy $k$.

This aggregation allows us to assess the overall contribution of each proxy to the classifier's predictions. To facilitate comparison across proxies, we normalize the aggregated importance scores. Specifically, let $I(p)$ denote the total importance score for proxy $p$ (i.e., the sum of importance scores for its four associated features). The normalized importance for proxy $p$ is computed as:

$$\widetilde{I}(p) = \frac{I(p)}{\sum_{p' \in \mathcal{P}} I(p')}$$

where $\mathcal{P}$ is the set of all proxies. This yields a distribution over proxies, where higher values indicate greater influence on the classifier's decision.

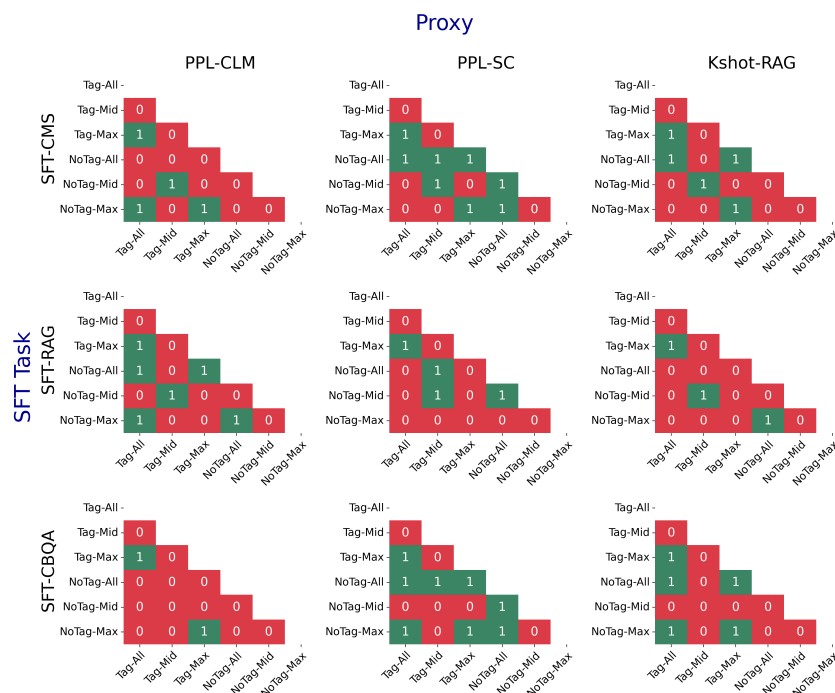

Figure 8: Predictive accuracy of PPL-CLM, PPL-SC, and Kshot-RAG in distinguishing the better-performing model between two LLMs with different length & filtering methods (other pre-trained configuration fixed). The naming follows the format of [Tagging]-[Length Filtering]. "Tag" and "NoTag" indicate whether domain tags are added. "All" keeps all examples, "Mid" keeps samples with lengths in the 25–75% quantile range, and "Max" keeps the longest 25% of examples.

## F    PROMPTS

The exampled prompts used for Kshot-CMS, Kshot-RAG, and Kshot-CBQA tasks are shown in Figure 9, Figure 10 and Figure 11 respectively.

## G    SUPERVISED FINETUNED, PERPLEXITY AND KSHOT RESULTS OF LLMS

The all supervised fine-tuned, perplexity and Kshot-learning results are detailed in Table 8.

|  | SFT-CMS | SFT-RAG | SFT-CBQA |
|---|---|---|---|
| **Conventional Perplexity** | | | |
| PPL-CLM | $.306\pm_{.081}$ | $.366\pm_{.060}$ | $.331\pm_{.054}$ |
| **Individual and Combined Proxies** | | | |
| Kshot-RAG | $.687\pm_{.073}$ | $.724\pm_{.047}$ | $.683\pm_{.077}$ |
| Combine Five Proxies | $.612\pm_{.055}$ | $.585\pm_{.051}$ | $.540\pm_{.104}$ |
| **Learning To Compare** | | | |
| **Train and Evaluate on the same task** | | | |
| Logistic Regression | $.738\pm_{.044}$ | $.688\pm_{.054}$ | $.624\pm_{.087}$ |
| Neural Networks | $\mathbf{.778}\pm_{\mathbf{.056}}$ | $.691\pm_{.055}$ | $.673\pm_{.071}$ |
| LightGBM | $.753\pm_{.054}$ | $\mathbf{.727}\pm_{.039}$ | $\mathbf{.753}\pm_{\mathbf{.060}}$ |
| **Train on SRC task** | | | |
| *Logistic Regresion* | | | |
| SFT-CMS (Src) | $.738\pm_{.044}$ | $.669\pm_{.059}$ | $.636\pm_{.060}$ |
| SFT-RAG (Src) | $.724\pm_{.074}$ | $.688\pm_{.054}$ | $.641\pm_{.079}$ |
| SFT-CBQA (SRC) | $.708\pm_{.069}$ | $.680\pm_{.049}$ | $.624\pm_{.087}$ |
| *Neural Networks* | | | |
| SFT-CMS (Src) | $.778\pm_{.056}$ | $.706\pm_{.060}$ | $0.683\pm_{.062}$ |
| SFT-RAG (Src) | $.742\pm_{.073}$ | $.691\pm_{.055}$ | $0.667\pm_{.075}$ |
| SFT-CBQA (Src) | $.748\pm_{.067}$ | $.695\pm_{.059}$ | $.673\pm_{.071}$ |
| *LightGBM* | | | |
| SFT-CMS (Src) | $.753\pm_{.054}$ | $.712\pm_{.054}$ | $.707\pm_{.057}$ |
| SFT-RAG (Src) | $.734\pm_{.047}$ | $.727\pm_{.039}$ | $.717\pm_{.071}$ |
| SFT-CBQA (Src) | $.734\pm_{.052}$ | $.718\pm_{.050}$ | $.753\pm_{.060}$ |

Table 7: Performance comparison of unsupervised baselines and supervised classifiers (Logistic Regression, Neural Networks, LightGBM) for predicting SFT-CMS, SFT-RAG, and SFT-CBQA. Results are reported as mean accuracy $\pm$ standard deviation over 20 runs.

You are an expert in commonsense reasoning tasks.
// five in-context examples in total.
Question: do iran and afghanistan speak the same language
Answer: True
...
Question: does canada's worst driver lose their license
Answer: No
Question: does canada's worst driver lose their license
Answer:

Figure 9: Prompt used for Kshot-CMS

You are an expert in question answering. I am going to give you five example triples of context, question and answer, in which the context may or may not be relevant to the question. The examples will be written.

// five in-context examples in total.
Context: <Retrieved documents>
Question: who sang the original blinded by the light
Answer: Bruce Springsteen
...
Context: <Retrieved documents>
Question: who played vincent in nanny mcphee and the big bang
Answer: Oscar Steer

Context: <Retrieved documents>
Question: how many episodes are there in dragon ball z
Answer:

Figure 10: Prompt used for Kshot-RAG. For each question, we retrieve the top-1 document as context using the Gecko-1B retriever Lee et al. (2024b).

You are an expert in question answering. I am going to give you five example of question-answer pairs as the in-context examples first. Your task is to generate a answer given a question.

// five in-context examples in total.
Question: the first life forms to appear on earth were
Answer: putative fossilized microorganisms
...
Question: who made the beavis and butthead theme song
Answer: Mike Judge

Question: what network is showing the monday night football game
Answer:

Figure 11: Prompt used for Kshot-CBQA.

| Model ID | Performance after Supervised Fine-tuning | | | Individual Proxies from Pre-Training | | | | |
| | SFT-CMS | SFT-RAG | SFT-CBQA | PPL-CLM | PPL-SC | Kshot-CMS | Kshot-RAG | Kshot-CBQA |
|---|---|---|---|---|---|---|---|---|
| 1 | 69.800 | 47.275 | 35.600 | 0.395 | 0.089 | 61.560 | 34.990 | 20.390 |
| 2 | 70.980 | 47.600 | 36.350 | 0.394 | 0.094 | 61.660 | 33.130 | 20.130 |
| 3 | 70.520 | 47.850 | 36.000 | 0.391 | 0.087 | 60.680 | 21.230 | 19.950 |
| 4 | 70.900 | 48.425 | 0.150 | 0.389 | 0.092 | 61.100 | 34.011 | 0.121 |
| 5 | 70.900 | 48.375 | 38.550 | 0.388 | 0.079 | 55.000 | 39.072 | 19.315 |
| 6 | 73.560 | 48.200 | 36.950 | 0.377 | 0.141 | 59.780 | 35.980 | 18.280 |
| 7 | 70.260 | 47.900 | 37.350 | 0.385 | 0.131 | 60.300 | 36.500 | 17.410 |
| 8 | 74.560 | 48.600 | 38.250 | 0.360 | 0.143 | 58.420 | 35.300 | 17.810 |
| 9 | 75.200 | 48.600 | 38.300 | 0.331 | 0.141 | 56.920 | 42.692 | 19.221 |
| 10 | 75.360 | 48.725 | 37.750 | 0.306 | 0.140 | 56.460 | 42.494 | 18.945 |
| 11 | 70.000 | 47.750 | 36.250 | 0.394 | 0.096 | 61.960 | 37.710 | 21.090 |
| 12 | 70.420 | 47.675 | 36.000 | 0.387 | 0.097 | 61.480 | 37.300 | 19.440 |
| 13 | 72.160 | 48.125 | 37.800 | 0.387 | 0.102 | 61.980 | 37.900 | 20.260 |
| 14 | 73.240 | 48.475 | 38.250 | 0.386 | 0.104 | 62.240 | 42.300 | 19.177 |
| 15 | 73.560 | 48.925 | 38.750 | 0.382 | 0.094 | 62.240 | 43.003 | 19.422 |
| 16 | 70.440 | 47.725 | 35.600 | 0.395 | 0.129 | 61.560 | 36.800 | 20.350 |
| 17 | 71.620 | 48.000 | 37.500 | 0.392 | 0.132 | 61.480 | 36.810 | 20.200 |
| 18 | 72.980 | 48.650 | 37.900 | 0.388 | 0.143 | 61.480 | 36.490 | 19.860 |
| 19 | 72.940 | 48.650 | 38.450 | 0.385 | 0.143 | 61.180 | 42.789 | 19.297 |
| 20 | 73.420 | 48.825 | 38.900 | 0.382 | 0.143 | 61.620 | 43.306 | 19.522 |
| 21 | 73.140 | 47.150 | 34.900 | 0.394 | 0.170 | 61.940 | 37.100 | 20.780 |
| 22 | 70.540 | 46.775 | 36.900 | 0.376 | 0.153 | 59.500 | 34.810 | 15.950 |
| 23 | 74.200 | 48.350 | 38.050 | 0.383 | 0.178 | 61.420 | 37.760 | 20.610 |
| 24 | 75.140 | 48.825 | 38.400 | 0.378 | 0.172 | 61.200 | 42.933 | 19.286 |
| 25 | 75.340 | 49.025 | 39.100 | 0.375 | 0.173 | 61.700 | 42.931 | 19.637 |
| 26 | 68.720 | 47.150 | 35.500 | 0.386 | 0.129 | 61.100 | 36.380 | 18.290 |
| 27 | 69.760 | 46.600 | 35.750 | 0.378 | 0.130 | 60.180 | 35.740 | 17.170 |
| 28 | 73.000 | 48.425 | 37.900 | 0.386 | 0.131 | 61.660 | 37.950 | 21.610 |
| 29 | 73.840 | 48.625 | 38.800 | 0.382 | 0.134 | 61.600 | 42.658 | 19.467 |
| 30 | 74.340 | 48.675 | 39.050 | 0.379 | 0.133 | 61.820 | 42.700 | 19.592 |
| 31 | 70.400 | 47.425 | 35.900 | 0.395 | 0.130 | 61.780 | 37.470 | 20.970 |
| 32 | 71.540 | 48.100 | 37.300 | 0.393 | 0.125 | 62.180 | 37.690 | 21.700 |
| 33 | 72.900 | 47.875 | 35.850 | 0.390 | 0.127 | 62.080 | 37.710 | 21.080 |
| 34 | 72.820 | 48.650 | 38.800 | 0.388 | 0.130 | 62.120 | 42.775 | 19.465 |
| 35 | 73.640 | 48.600 | 38.450 | 0.385 | 0.129 | 61.560 | 42.711 | 19.290 |
| 36 | 71.620 | 47.625 | 37.700 | 0.364 | 0.102 | 61.680 | 31.760 | 20.280 |
| 37 | 71.700 | 47.900 | 37.250 | 0.373 | 0.102 | 61.640 | 33.080 | 19.940 |
| 38 | 70.200 | 47.650 | 37.700 | 0.374 | 0.096 | 51.580 | 11.330 | 1.230 |
| 39 | 71.080 | 47.825 | 37.550 | 0.387 | 0.110 | 60.800 | 33.860 | 20.290 |
| 40 | 71.480 | 48.000 | 37.850 | 0.389 | 0.107 | 60.720 | 33.170 | 19.250 |
| 41 | 72.400 | 48.000 | 37.800 | 0.360 | 0.101 | 61.880 | 37.180 | 19.720 |
| 42 | 72.300 | 48.125 | 37.300 | 0.368 | 0.103 | 62.200 | 37.610 | 19.390 |
| 43 | 72.360 | 48.100 | 37.350 | 0.368 | 0.104 | 62.180 | 37.370 | 20.040 |
| 44 | 72.800 | 48.350 | 37.550 | 0.382 | 0.111 | 62.300 | 37.660 | 20.320 |
| 45 | 72.480 | 47.825 | 38.000 | 0.383 | 0.111 | 61.560 | 37.870 | 20.860 |
| 46 | 72.220 | 47.900 | 37.650 | 0.380 | 0.104 | 61.860 | 26.500 | 20.160 |
| 47 | 72.040 | 47.575 | 37.300 | 0.387 | 0.106 | 61.120 | 32.380 | 20.200 |
| 48 | 71.800 | 47.325 | 37.350 | 0.386 | 0.107 | 61.160 | 33.210 | 18.540 |
| 49 | 72.220 | 47.900 | 37.650 | 0.380 | 0.104 | 61.860 | 26.500 | 20.160 |
| 50 | 72.040 | 47.575 | 37.300 | 0.387 | 0.106 | 61.120 | 32.380 | 20.200 |

Table 8: SFT, perplexity and kshot performance for all pretrained LLMs.

