# OpenReview forum: "When Scale is Fixed: Revisiting Pre-training Indicators for LLM Fine-tuning Performance"
_ICLR.cc/2026/Conference — Submitted to ICLR 2026_

### Official Review · Reviewer_wgMi · 2025-10-18

**Soundness:** 3
**Presentation:** 3
**Contribution:** 2
**Rating:** 6
**Confidence:** 3

**Summary:**

This paper tackles an underexplored problem: predicting fine-tuning performance of large language models (LLMs) when model scale is fixed. Prior work largely focuses on scaling laws—how performance changes with size or compute—but this paper isolates pre-training effects (data, objectives, filtering) at a constant parameter count (1B). The authors cast the task as pairwise prediction: given two pre-trained models, which will perform better after supervised fine-tuning (SFT)? Using 50 systematically varied 1B models, they evaluate existing pre-training indicators (e.g., perplexity) and propose new unsupervised and supervised proxies. The main contribution is a Learning-to-Compare (LTC) classifier that integrates multiple pre-training metrics and cuts prediction error by >50% relative to perplexity.

**Strengths:**

1. The paper makes a clear conceptual departure from scaling-law analyses by focusing on within-scale predictive indicators of fine-tuning performance. This framing is both original and practically significant, as it directly addresses the common real-world scenario in which multiple pre-training checkpoints of identical size must be compared efficiently.
2. The study’s formulation of the evaluation problem as pairwise classification over more than 1,000 model pairs is statistically sound and well-justified for the task of model ranking.
3. The experimental results compellingly demonstrate that conventional perplexity is a poor predictor of downstream fine-tuning performance within a fixed model scale (accuracy ≤ 0.33).

**Weaknesses:**

1.  Details of exp setup are missing. What LLM are used in this work? What are their parameter settings? If I understand correctly, all results are on 1B-parameter models. Without even partial scaling evidence (e.g., 3B subset), it’s uncertain if the same patterns hold at practical large-model scales (7B–70B).
2. The paper claims systematic variations in data composition and filtering but doesn’t analyze which factor contributes most to proxy predictability.
3. Although cross-validation and 20 random splits are used, the dataset of 50 models is small.
4. The feature-importance plot shows Kshot-RAG dominates, but it’s unclear whether the LTC gains come mostly from simple nonlinear reweighting. An ablation of proxy subsets could clarify if supervised fusion is truly synergistic or just re-scaling.

**Questions:**

1. Provide sensitivity analysis isolating pre-training objective vs. data composition.
2. Include scaling-up validation on at least 3B or 7B checkpoints, even partial.
3. Detail the statistical control to prevent leakage or overfitting in LTC training.
4. Clarify proxy combination mechanisms and perform ablation across subsets.

---

> ### Author Response · Authors · 2025-11-19
>
> **Q1**:
> Details of exp setup are missing. What LLM are used in this work? What are their parameter settings?
>
> **A1**:
> Thank you for the suggestion. We utilize a standard Decoder-only Transformer architecture, implemented using the JAX/T5X framework. Our backbone configuration follows the modern architectural modifications detailed in Chowdhery et al. (2022) and Garcia et al. (2023). For all experiments in this work, we adopt the 1B-parameter  model settings.
>
> **References:**
> * Chowdhery, A., et al. "PaLM: Scaling Language Modeling with Pathways." *JMLR*, 2023.
> * Garcia, X., et al. "The Unreasonable Effectiveness of Few-Shot Learning for Machine Translation." *ICML*, 2023.
>
> **Q2**: Provide sensitivity analysis isolating pre-training objective vs. data composition.
>
> **A2**: We actually conducted this isolation analysis in Section 3 ("A predictive power case study...") and showed in Figures 2, 6, and 7.
> We grouped model pairs to ensure they differed only in one dimension while keeping other factors fixed:
> 1. **Isolating Pre-training Objectives (Figure 2) & Data Composition (Figure 6):** When comparing models that differ only in their objective or only in their data mixture, we observe a consistent trend: `PPL-SC` and `Kshot-RAG` significantly outperform `PPL-CLM`. This demonstrates that `PPL-SC` and `Kshot-RAG` are robust indicators for capturing performance gaps caused by both algorithmic changes and data distribution shifts, whereas standard perplexity (`PPL-CLM`) struggles in these isolated settings.
>
> 2. **Isolating Data Filtering (Figure 7):** In contrast, when models differ only in filtering strategies, all three proxies struggle to distinguish the better model. This suggests that the performance differences caused by filtering variations are more subtle and harder to capture with the current set of proxies.
>
> Comparing these isolated factors reveals that the pre-training objective and data composition are the dominant factors contributing to proxy predictability. The signals generated by objective differences are much stronger and more effectively captured by `PPL-SC` and `Kshot-RAG` than those from data filtering. We will make these comparative conclusions explicit in the Discussion section of the final revision.
>
> **Q3**:
> Include scaling-up validation on at least 3B or 7B checkpoints, even partial.
>
> **A3**:
> Thanks for your suggestion. We acknowledge the importance of validating findings at larger scales. However, our experimental design deliberately prioritized **sample size ( $N=50$ models)** over parameter size to ensure the statistical rigor of our pairwise ranking formulation.
>
> We chose the 1B scale because it offers the optimal trade-off between representational capacity and computational efficiency. This allowed us to conduct systematic comparisons across a large population of pre-training variants, generating enough model pairs for training and testing. Training 50+ models at the 7B scale to maintain this statistical density is computationally prohibitive for this study.
>
> Regarding "partial" validation: analyzing only a few 3B or 7B checkpoints would yield a sample size too small to form a statistically significant test set for the pairwise classifier, potentially introducing noise rather than clarity. We posit that the 1B scale serves as a robust testbed for establishing the methodology of indicator fusion. Since the correlation between proxies and downstream performance is primarily driven by pre-training objectives and data configurations rather than raw scale, we expect these methodological findings to generalize to larger models.
>
> We will add a discussion on these computational trade-offs and the scope of generalization in the final revision.
>
> **Q4**:
> Detail the statistical control to prevent leakage or overfitting in LTC training.
>
> **A4**:
> To ensure statistical rigor and prevent leakage, we implemented two strict controls:
>
> 1.  **Model-Level Splitting (Preventing Leakage):** We performed the train/test split at the **model level**, not the pair level. We randomly sample 60% of models in training, and the remaining 40% in testing sets. This means the set of models used for testing was completely disjoint from those used for training. By doing this, we ensure the classifier predicts performance based on learned feature relationships, rather than memorizing the identity of specific models seen during training.
>
> 2.  **Robustness via Repeated Experiments (Mitigating Overfitting):** We repeated the entire experimental pipeline **20 times using different random seeds** for the train/test splits. Reporting the average accuracy across these 20 independent runs ensures our results are stable and not due to overfitting to a specific, favorable split.

---

> ### Author Response · Authors · 2025-11-19
>
> **Q5**: Clarify proxy combination mechanisms and perform ablation across subsets.
>
> **A5**:
> We employed a LightGBM classifier to learn the pairwise ranking of model performance. To systematically analyze feature contributions and interactions, we conducted an exhaustive ablation study by enumerating all possible combinations of our 5 proxy features, varying the subset size from $k=1$ to $k=5$.  For each subset size k, we selected the combination achieving the highest accuracy and visualized these results in a Figure.
> To ensure the statistical robustness of our findings, each feature combination was evaluated across 20 distinct random seeds, and we report the average accuracy. And the Figure https://ibb.co/nKr3pSk illustrates the average pairwise accuracy for each target task as the number of features in the combination increases from $k=1$ to $k=5$.
>
> **Observations**: At the $k=1$ baseline, we found that Pile-SC was the most effective single metric for predicting SFT-CMS performance (Acc. $\approx 0.61$), while Kshot-RAG was the dominant single predictor for both SFT-RAG and SFT-CBQA (Acc. $\approx 0.58$ and $\approx 0.61$, respectively). Crucially, combining just two features ($k=2$) resulted in substantial performance improvements over these baselines: SFT-CMS accuracy improved by 11.67%, SFT-RAG by 12.06%, and SFT-CBQA by 9.49%. The overall performance generally trended upwards as the feature combination size increased
>
> **Conclusion**: These findings provide strong evidence for feature synergy, demonstrating that the model fuses complementary information ($k=2$) to achieve predictive accuracy significantly beyond what is possible with simple re-scaling of a single dominant feature ($k=1$). And overall performance increases when the feature combination size increases
>
> We have added this experiment and the corresponding analyses to Section 8.3 of the newly uploaded revised paper.

---

### Official Review · Reviewer_sUmD · 2025-11-01

**Soundness:** 2
**Presentation:** 3
**Contribution:** 3
**Rating:** 6
**Confidence:** 2

**Summary:**

This paper studies whether pre-training indicators can predict fine-tuning performance when model scale is held constant. While perplexity is widely used as a progress indicator during pre-training and is known to correlate with downstream performance across scaling laws, its predictive power at a fixed model size remains unclear. The authors systematically pre-train 50 variants of a 1B-parameter language model, varying pre-training objectives, data mixture weighting, and tagging. They then fine-tune all variants across three downstream categories. Results show that standard perplexity correlates poorly with fine-tuning outcomes. In contrast, proxies such as span-corruption perplexity and k-shot performance better predict downstream results.

**Strengths:**

This paper addresses an important and underexplored question: whether pre-training indicators can predict fine-tuning performance when model scale is fixed. The authors formulate this as a pairwise prediction problem and conduct a systematic empirical study across 50 variants of a 1B-parameter LLM. The experimental results provide evidence that conventional perplexity is an unreliable predictor for post-SFT performance, while span corruption perplexity and few-shot evaluation metrics offer significantly better predictive power. The proposed supervised Learning-to-Compare framework further integrates multiple proxies to enhance prediction accuracy and demonstrates cross-task generalization.

**Weaknesses:**

1. While the experiments on 1B-parameter models are internally consistent, it would strengthen the paper to include one or two experiments verifying whether the observed trends extrapolate to larger, more realistic LLMs.

2. The evaluation currently covers only three SFT tasks (CMS, RAG, CBQA); it is unclear whether the proposed proxies generalize to broader SFT tasks.

3. The paper reports pairwise prediction accuracy as the main evaluation metric; it would be helpful to also present ranking-based measures.

**Questions:**

Please refer to weaknesses.

---

> ### Author Response · Authors · 2025-11-19
>
> **Q1**: While the experiments on 1B-parameter models are internally consistent, it would strengthen the paper to include one or two experiments verifying whether the observed trends extrapolate to larger, more realistic LLMs.
>
> **A1**: Thanks for your suggestion. We acknowledge the importance of validating findings at larger scales. However, our experimental design deliberately prioritized **sample size ( $N=50$ models)** over parameter size to ensure the statistical rigor of our pairwise ranking formulation.
>
> We chose the 1B scale because it offers the optimal trade-off between representational capacity and computational efficiency. This allowed us to conduct systematic comparisons across a large population of pre-training variants, generating enough model pairs for training and testing. Training 50+ models at the 7B~70B scales to maintain this statistical density is computationally prohibitive for this study.
>
> Regarding small-scale validation: analyzing only a few 7B~70B checkpoints would yield a sample size too small to form a statistically significant test set for the pairwise classifier, potentially introducing noise rather than clarity. We posit that the 1B scale serves as a robust testbed for establishing the methodology of indicator fusion. Since the correlation between proxies and downstream performance is primarily driven by pre-training objectives and data configurations rather than raw scale, we expect these methodological findings to generalize to larger models.
>
>
> **Q2**: The evaluation currently covers only three SFT tasks (CMS, RAG, CBQA); it is unclear whether the proposed proxies generalize to broader SFT tasks.
>
> **A2**:
> CMS, RAG, and CBQA cover the major use cases of LLMs, including commonsense and reasoning ability (CMS), retrieval-based generation (RAG), and factual recall (CBQA). Each group also includes multiple datasets (e.g., BoolQ, PIQA, HotpotQA, NQ, TriviaQA), providing broad coverage of diverse SFT scenarios. In the future, we will consider including reasoning-intensive tasks such as math problem solving and code generation.
>
>
> **Q3**:
> The paper reports pairwise prediction accuracy as the main evaluation metric; it would be helpful to also present ranking-based measures.
>
> **A3**:
> Actually, we have already included a ranking-based evaluation in Section 5.2 (“Recall the Best Model from a Small Candidate Set”) and Figure 5.
> Specifically, we aggregate pairwise comparison results using Borda Count scoring and report Top-1 and Top-5 recall, which directly measure our model’s ability to rank and identify the best pre-trained LLMs within candidate sets. As shown in Figure 5, the Learning-to-Compare (LTC) method consistently outperforms both Kshot-RAG and PPL-CLM, achieving perfect Top-1 recall across SFT-CMS, SFT-RAG, and SFT-CBQA tasks

---

### Official Review · Reviewer_E6RE · 2025-11-01

**Soundness:** 4
**Presentation:** 3
**Contribution:** 3
**Rating:** 4
**Confidence:** 3

**Summary:**

This paper investigates how to predict the fine-tuning (SFT) performance of large language models when model size and compute are fixed. Using 50 one-billion-parameter LLMs trained with different pre-training objectives and data mixtures, the authors show that standard perplexity fails to predict post-SFT performance. They introduce alternative proxies, like span-corruption perplexity and few-shot (k-shot) scores, and a learning-to-compare (LTC) framework that combines these signals via supervised classification. The LTC approach improves pairwise prediction accuracy by over 50% versus perplexity and reliably identifies the best checkpoints, offering a practical method for efficient model selection during pre-training.

**Strengths:**

1. The paper conducts comprehensive evaluation using multiple SFT tasks and metrics (CMS, RAG, CBQA). It tests both unsupervised proxies and supervised meta-predictors. It also provides systematic experiment design that include 50 model variants with consistent size and compute budgets.
2. The paper offers insightful empirical findings, showing strong evidence that perplexity is not reliable for fixed-size model selection.
3. I also like the pairwise evaluation formulation, which is simple and interpretable.

**Weaknesses:**

1. Experiments confined to 1B-parameter models; unclear if findings hold for larger LLMs (10B–70B) which is commonly used nowadays. While I understand this could because of the computation cost, a small-scale study would be interesting in this parameter size range.
2. The paper provides largely empirical insights and there are limited theoretical or mechanistic analysis on why certain proxies work better. It will be great if theoretical reasoning or model can be provided to shed some light.
3. The “learning-to-compare” framework is a straightforward application of supervised ranking. While it’s effective, it’s not conceptually novel.

**Questions:**

See weakness.

---

> ### Author Response · Authors · 2025-11-19
>
> **Q1:** Experiments confined to 1B-parameter models; unclear if findings hold for larger LLMs (10B–70B) which is commonly used nowadays. While I understand this could because of the computation cost, a small-scale study would be interesting in this parameter size range.
>
> **A1:**
> Thanks for your suggestion. We acknowledge the importance of validating findings at larger scales. However, our experimental design deliberately prioritized **sample size ( $N=50$ models)** over parameter size to ensure the statistical rigor of our pairwise ranking formulation.
>
> We chose the 1B scale because it offers the optimal trade-off between representational capacity and computational efficiency. This allowed us to conduct systematic comparisons across a large population of pre-training variants, generating enough model pairs for training and testing. Training 50+ models at the 10B~70B scales to maintain this statistical density is computationally prohibitive for this study.
>
> Regarding small-scale validation: analyzing only a few 10B or 70B checkpoints would yield a sample size too small to form a statistically significant test set for the pairwise classifier, potentially introducing noise rather than clarity. We posit that the 1B scale serves as a robust testbed for establishing the methodology of indicator fusion. Since the correlation between proxies and downstream performance is primarily driven by pre-training objectives and data configurations rather than raw scale, we expect these methodological findings to generalize to larger models.
>
>
> **Q2:** The paper provides largely empirical insights and there are limited theoretical or mechanistic analysis on why certain proxies work better. It will be great if theoretical reasoning or model can be provided to shed some light.
>
> **A2:**
> Thank you for the suggestion. While our work is primarily empirical, we offer the following reasoning for why PPL-SC and Kshot-RAG outperform the standard PPL-CLM.
>
> Regarding PPL-SC vs. PPL-CLM, our findings align with observations in Tay et al. (2022) [1]. The authors found that including the S-denoiser (SC) in pre-training objectives specifically favors downstream fine-tuning performance, whereas standard CLM tends to favor few-shot performance. This suggests that PPL-SC effectively captures the sequential, conditional generation dynamics required for SFT, which standard perplexity (PPL-CLM) may miss.
>
> Regarding Kshot-RAG vs. PPL-CLM, we hypothesize this advantage stems from the difference between *memorization* and *information synthesis*. Standard perplexity largely measures how well a model memorizes pre-training data. However, SFT requires the model to process input instructions and synthesize a response. Kshot-RAG directly tests this capability to utilize context for reasoning, making it a stronger predictor of instruction-following potential than simple data memorization.
>
> We plan to investigate these theoretical connections more deeply in our future work.
>
> **References:**
> [1] Tay, Y., et al. "UL2: Unifying Language Learning Paradigms." *International Conference on Learning Representations (ICLR)*, 2022.
>
>
> **Q3:**
> The “learning-to-compare” framework is a straightforward application of supervised ranking. While it’s effective, it’s not conceptually novel.
>
> **A3:**
> The novelty of our work does not lie in proposing a new algorithm, but rather in the systematic empirical finding that PPL-CLM is not a reliable indicator of downstream task performance when model scale and training compute are fixed. And propose several solution for that.
>
> While our Learning-to-Compare (LTC) framework indeed draws inspiration from Learning-to-Rank approaches in information retrieval, its purpose here is not algorithmic innovation, but practical validation: showing that one can predict downstream fine-tuning performance using only metrics available at the pre-training stage (e.g., PPL-SC, Kshot-RAG, etc.) with minimal additional computation.
>
> This contributes a lightweight and interpretable way to track and compare pre-trained models, enabling early-stage performance monitoring and selection before expensive fine-tuning

---

### Official Review · Reviewer_S9nM · 2025-11-01

**Soundness:** 1
**Presentation:** 2
**Contribution:** 1
**Rating:** 2
**Confidence:** 4

**Summary:**

The paper addresses the problem of predicting which of two checkpoints will perform better after supervised fine-tuning (SFT). It evaluates the discrimination accuracy of two types of proxies: perplexity on the Pile and few-shot performance. These proxies are then used as features to train a classifier, which achieves higher classification accuracy than any single proxy.

**Strengths:**

* The problem addressed is both interesting and important.
* The experimental setup is described in detail, and the data used for analysis are provided in Table 6. This transparency increases confidence in the replicability of the results.
* The presentation is generally clear and easy to follow.

**Weaknesses:**

A core claim of the paper is that perplexity is a misleading indicator of performance after SFT. I believe that not enough evidence is put forward to substantiate this claim. First, only perplexity on the Pile dev set is considered. Authors should explore at least a few other PPL indicators; for example, PPL on a test set drawn iid from DC-0, PPL on the CC split of SlimPajama, PPL on the Wiki split of SlimPajama, and mean PPL on the train sets of the benchmarks considered. Relative to training models, these experiments should be rather cheap to run.

Secondly, the results imply that higher PPL is indicative of better performance (e.g., see Table 1 top row, this rule would lead to 100% - 33% = 67% accuracy for SFT-CMS). This is highly surprising and extremely counterintuitive (if it holds beyond Pile dev), so much so that it deserves additional explanation of (1) why this might arise in the considered setup and (2) whether we can we expect similar results for more realistic setups. For example, the architecture search community compares architectures almost exclusively based on PPL evaluations, should we sound the alarm that they are hill climbing in the diametrically opposite direction? I am inclined to think not.

The work also has a serious flaw in the analysis: it does not account for statistical significance of either (1) differences in benchmark performance, or (2) differences in the pre-training proxy indicators. For example, in Table 6 SFT-CMS, model 18 has a mean accuracy of 72.98, whereas model 19 has a mean accuracy of 72.94. These two SFT models should be considered equivalent. Authors should repeat their analysis taking into account statistical significance in model performance and the individual proxies.

Authors find that the most accurate proxy is Kshot-RAG. Looking at Table 6, I would argue that this is simply because it is the one that gives the most resolution in ranking models (e.g., and not because Kshot-RAG is otherwise particularly indicative of downstream performance). I encourage authors to make the following simple analysis: among the 12 benchmarks considered, take the top 4 that give the most resolution in ranking models (e.g., those benchmarks with largest mean difference in accuracy between contiguous model ranks relative to mean accuracy across all ranks). Then check how good of an indicator mean performance across these 4 models is.

**Questions:**

* For QA benchmarks, are models evaluated (1) by choosing the answer choice with lowest PPL given the question, (2) MMLU-style (e.g., A, B, C, D), or (3) by generating from the model and using some form of answer matching? I would recommend the former as this should give the most resolution in the benchmark evals.

---

> ### Author Response · Authors · 2025-11-19
>
> **Q1:** Additional perplexity evaluations (SlimPajama-CC, SlimPajama-Wiki, benchmark PPL) are missing.
>
> **A1:** Thank you for this thoughtful comment. We agree that evaluating perplexity only on the Pile dev set is insufficient to fully assess whether perplexity is a reliable predictor of post-SFT downstream performance.
>
> To address this concern, we have conducted all additional PPL-based proxy evaluations requested by the reviewer, including:
>
> - SlimPajama-Wiki perplexity
> - SlimPajama-CC perplexity
> - Mean perplexity over the benchmark training sets (“Benchmarks PPL-CLM”)
> - (Plus the original) Pile-dev perplexity (PPL-CLM) and Kshot-RAG for reference.
>
> We evaluate each proxy using the pairwise classification accuracy metric. The results are summarized below:
>
> | Proxy Indicator         |   SFT-CBQA |   SFT-CMS |   SFT-RAG |   Mean Accuracy |
> |-------------------------|-----------:|----------:|----------:|----------------:|
> | Kshot-RAG               | 0.7064     | 0.6972    | 0.7685    | 0.7240          |
> | Benchmarks PPL-CLM      | 0.4607     | 0.4133    | 0.4863    | 0.4534          |
> | SlimPajama-Wiki PPL     | 0.4450     | 0.3813    | 0.4664    | 0.4309          |
> | PPL-CLM (Pile-dev)      | 0.3350     | 0.3101    | 0.3635    | 0.3362          |
> | SlimPajama-CC PPL       | 0.3110     | 0.2881    | 0.3212    | 0.3067          |
>
> Key findings based on these expanded experiments:
>
> 1. All perplexity-based proxies show substantially lower pairwise accuracy compared to k-Shot RAG evaluation (mean accuracy 0.30–0.46 for PPL vs. 0.72 for Kshot-RAG).
> 2. Even in the most favorable setting (“Benchmarks PPL-CLM”, i.e., PPL computed directly on the training distributions of the downstream tasks), perplexity explains only ~45% of pairwise preferences.
> 3. These results confirm our central claim:
>    Perplexity is not a reliable indicator of relative downstream performance after SFT, even when evaluated on carefully chosen or domain-matched datasets.
>
> We have added this experiment and the corresponding analyses to Section 8 of the revised paper.
>
>
> **Q2:** The results seem to suggest that higher perplexity corresponds to better downstream performance, which is counterintuitive. Why does this happen in our setup? Would it occur in more realistic scenarios, and does it imply that architecture search methods— which rely heavily on perplexity ranking—are optimizing in the wrong direction?
>
> **A2:** Our main contribution is that we focus on the pre-training stage, where, given identical model size and training compute, CLM perplexity fails to serve as a reliable proxy for downstream performance.
> In contrast, Neural Architecture Search (NAS) approaches, such as LLM-Pruner [1], study post-training pruning and efficiency optimization after the models have already been pre-trained.
> Hence, the two areas address orthogonal questions—our work concerns understanding pre-training dynamics and metric predictiveness, whereas theirs focuses on architectural compression and deployment efficiency—and should not be compared directly.
>
> [1] LLM-Pruner: On the Structural Pruning of Large Language Models. Xinyin Ma, Gongfan Fang, Xinchao Wang. NeurPIS 2023.
>
>
> **Q3:** Does not account for the statistical significance of either (1) differences in benchmark performance, or (2) differences in the pre-training proxy indicators.
>
> **A3:** We agree that tiny gaps like 72.98 vs 72.94 are not meaningful and likely come from SFT randomness.
> This issue is already handled in Section 5.1 and Figure 4. Instead of comparing raw scores, we grouped all model pairs into five bins based on their absolute accuracy difference (0–20%, 20–40%, …, 80–100%).
> Pairs with the smallest differences fall into the first quantile, and those with the largest differences fall into the fifth quantile.
> As shown in Figure 4, the Kshot-RAG’s predictive accuracy increases when the SFT performance difference increases, while PPL-CLM does not.
> This demonstrates that after controlling for noisy pairs, our main conclusion still holds.
>
> Additionally, we also accounted for randomness in the pairwise accuracy comparison in Table 2. We randomly split the 50 models into 60% training and 40% testing, repeated this procedure 20 times, and measured the variance. After doing so, we still observed that Kshot-RAG significantly outperforms PPL-CLM.

---

> ### Author Response · Authors · 2025-11-19
>
> **Q4:** Is Kshot-RAG truly a better proxy, or does it only appear stronger because it provides higher ranking resolution (as seen in Table 6)? To verify this, one should select the four benchmarks with the highest resolution and check whether the mean performance over these four benchmarks serves as an equally strong indicator.
>
> **A4:** As suggested, we select the top-4 benchmarks with the highest resolution: CBQA_TriviaQA, CBQA_NQ, Winogrande, and OpenBookQA.
>
> We then computed the mean pairwise **accuracy** of each proxy following the same definition used in Table1.
>
> | Metric | CBQA_TriviaQA | CBQA_NQ | Winogrande | OpenBookQA | mean |
> | :--- | :--- | :--- | :--- | :--- | :--- |
> | **Kshot_RAG** | 0.7192 | 0.6841 | 0.6947 | 0.6873 | **0.6963** |
> | **Pile_SC** | 0.6024 | 0.5494 | 0.6424 | 0.6735 | 0.6169 |
> | **Kshot_CMS** | 0.5159 | 0.5910 | 0.5706 | 0.6016 | 0.5698 |
> | **KShot_CBQA** | 0.4645 | 0.5371 | 0.4661 | 0.4980 | 0.4914 |
> | **Pile_CLM** | 0.3404 | 0.4555 | 0.2931 | 0.3706 | 0.3649 |
>
> The trend is consistent with Table 1:
>
> 1. Kshot-RAG achieves the **highest mean accuracy** among individual proxies.
> 2. Pile-CLM (perplexity) shows the **lowest accuracy**.
>
> This consistent trend in both the full set and the high-resolution subset shows that Kshot-RAG’s advantage is not due to benchmark resolution. Instead, it truly captures downstream performance better than other proxies.
>
> We have added the experiment and analysis in Section 8.2 of the newly uploaded revised paper.
>
>
> **Q5:** Which evaluation method is used for the QA benchmarks: (1) selecting the answer choice with the lowest PPL, (2) treating it in an MMLU-style multiple-choice format, or (3) generating an answer and matching it?
>
> **A5:** We adopt the first method: for each answer choice, we compute its perplexity (PPL) and select the choice with the lowest value.

---

### Meta-Review · Area_Chair_cTG7 · 2025-12-24

**Summary:**

This paper argues that perplexity is not necessarily a strong indicator of post-SFT performance. The paper performs a rigorous analysis of 50 1B parameter models to show limitations in perplexity and suggest an alternative learning-to-rank strategy.

Reviewer concerns were primarily along two axes: 1) ablating the method against alternatives (subsets and other ranking methods) to better understand the impact of the proposed policy; and 2) generalization towards different model scales. While the rebuttal has better clarified the impact of the policy, there remains a fundamental challenge in that all analysis is strictly limited to 1B models. This is an issue as the studied properties may not necessarily scale to smaller or larger models. While I acknowledge the rigor of the current study and the challenges of scaling it in full to larger models, I suggest that some attempt be made to demonstrate the scalability of the analysis, ideally with some small-size analysis at larger scale. As such, I cannot recommend acceptance.

**Reviewer Concerns:**

- Detailed analysis towards which factors (e.g., data composition, filtering) contribute to proxy predictability: The rebuttal confirms that this was performed.
- Missing ablations against alternatives (subsets and other ranking methods) to determine where the value from the LTC policy is being derived: Multiple reviewers suggested several alternatives. The rebuttal is reasonably comprehensive in addressing all of those either with new experiments or confirming the current results.
- Generalization towards larger models: The rebuttal argues that given the cost of large-scale training, the specific setup is within scope. Although I agree with the rebuttal that the current study is rigorous, I also agree with the reviewers that some variation in model scale is necessary, especially as it is known that model trends do not always behave similarly across scale. This could have been addressed either with some smaller experiments at large scale, or some experiments at smaller scale.
- Generalization towards other tasks: The rebuttal argues that the studied tasks are reasonably comprehensive and suggest remaining tasks to be left to future work. I agree that the approach is reasonably comprehensive.
- Demonstrating the weakness of perplexity as an estimator: The rebuttal includes new experiments evaluating PPL-based proxy evaluations to re-affirm their claim.
- Statistical significance of differences: The rebuttal clarifies that the paper does attempt to address statistical significance in a slightly roundabout way.

**Reviewer Scores:**

I do not believe that any of the reviewers would change their scores meaningfully. While the rebuttal is reasonably comprehensive, multiple reviewers remarked on the limitation of focusing on a single scale of model size. The rebuttal focused on the rigor of the current analysis and remarked on difficulty of scaling to larger settings. However, I do not believe this is satisfactory as some preliminary analysis (either at small scale for large models, or comprehensive for smaller models) could have mitigated these concerns.

---

### Decision · Program_Chairs · 2026-01-26

Reject